# Sublingual Sufentanil Tablet System (SSTS-Zalviso^®^) for Postoperative Analgesia after Orthopedic Surgery: A Retrospective Study

**DOI:** 10.3390/jcm11226864

**Published:** 2022-11-21

**Authors:** Andrea Angelini, Gian Mario Parise, Mariachiara Cerchiaro, Francesco Ambrosio, Paolo Navalesi, Pietro Ruggieri

**Affiliations:** 1Department of Orthopedics and Orthopedic Oncology, University of Padova, 35128 Padova, Italy; 2Institute of Anesthesiology and Critical Care, Department of Medicine-DIED, University of Padova, 35128 Padova, Italy

**Keywords:** sublingual sufentanil tablet system (SSTS–Zalviso^®^), total knee arthroplasty, retrospective study, postoperative pain, continuous femoral nerve block

## Abstract

Background: The aim of this study is to compare sublingual sufentanil and the administration device for its delivery (SSST-Zalviso^®^) with the traditional strategies used for the control of postoperative pain to establish if there is an actual benefit for the patient and healthcare personnel. Materials and Methods: A retrospective study was conducted to compare the efficacy of SSTS in the management of postoperative pain after orthopedic surgery between October 2018 and June 2020. We analyzed 50 patients who underwent a total knee arthroplasty (TKA). The control group consisted of 21 patients who underwent TKA and during the hospitalized recovery received a continuous femoral nerve block (cFNB). The statistical study was conducted with a level of significance *p* = 0.05 using “U” test, Mann–Whitney, to verify if patients had a better control of pain and fewer calls for rescue analgesia. Results: Patients involved in the study showed a significant reduction in pain intensity with the use of SSTS in the 24 h following surgery (*p* = 0.0568), also a drastic drop of the calls for rescue analgesia (*p* < 0.0001) reduces the number of calls for its control. Conclusions: This study demonstrates how SSTS might reduce pain intensity in the first 24 h after surgery and reduce the number of calls for its control, indicating better analgesic coverage and implying reduced interventions from healthcare personnel. This could allow a redistribution of resources and a reduction in the use of analgesic drugs in wards where the SSTS is used.

## 1. Introduction

Postoperative pain after orthopaedic procedures, such as total knee arthroplasty (TKA), is a major problem and represents a challenge. Pain control following TKA is a prerequisite for good outcomes because rehabilitation requires early mobilization and enhanced recovery [1]. TKA has become a fast-track approach, where early rehabilitation is a key factor in reducing morbidity and decreasing the length of hospital stay [2]. Despite the increasing knowledge about postoperative pain management and the implementation of new pain management techniques, an adequate protocol is still debated in the literature [3,4,5,6,7].

The number of total knee arthroplasty procedures has increased over the last 2–3 decades and is projected to increase even further due to population aging and longevity; however, these patients are affected by significant postoperative pain [8,9]. This pain, if uncontrolled, can hinder the recovery process by setting in motion-detrimental pathophysiological processes that increase the risk of postoperative complications. Ineffective pain control can result in low limb mobility, which can result in several medical morbidities, such as venous thrombosis, coronary ischemia, myocardial infarction, and pneumonia [10]. Inefficient pain control can also hinder mobilization and rehabilitation, disrupt sleep, cause cognitive dysfunction, and increase patient anxiety. Therefore, focusing on effective pain control among total knee arthroplasty patients is crucial, as it influences recovery time, cost of healthcare, and overall patient satisfaction [11,12]. The Enhanced Recovery After Surgery (ERAS) protocols make it possible to achieve these objectives thanks to a specific multidisciplinary approach. The advantages include the reduction in the average hospital stay and a low rate of complications, resulting in greater patient satisfaction [13,14]. Early mobilization represents one of the main aspects of ERAS and is closely linked to the use of short-acting anesthetic techniques aimed at minimizing side effects despite multimodal analgesic coverage [13,15]. There is wide debate on the best postoperative analgesia technique after TKA, with the choice of peripheral nerve blocks, epidural analgesia, and local infiltration analgesia (LIA) or (systemic) opioids. [1,16,17,18,19,20]. Sufentanil is a potent synthetic µ-opioid receptor specific agonist primarily used for intraoperative surgical analgesia. Sufentanil can be administered via sublingual route, demonstrating a good bioavailability (60%) and a prolonged duration of action compared to IV-Sufentanil, thanks to its high lipophilicity. For these reasons, sublingual sufentanil has recently been used for postoperative pain management and marketed as a pre-programmed, non-invasive handheld device (The Sufentanil Sublingual Tablet System (SSTS)—Zalviso^®^, Grünenthal GmbH, Aachen, Germany) [13,21,22]. This system combines the advantages of the non-IV route with self-administration (Figure 1).

The purpose of this study is to compare sublingual sufentanil and the administration device for its delivery (SSTS) with the traditional strategies used for the control of postoperative pain to establish if there is an actual benefit for the patient and healthcare personnel [23,24,25]. Therefore, we compared the efficacy of SSTS and continuous femoral nerve block (CFNB) in the postoperative pain control in patients treated with TKA. We propose that SSTS could show a better profile of pain control, tolerability, and patient and staff satisfaction, reducing the intensity of pain and side effects [25,26]. The primary objectives were to assess the efficacy of SSTS in controlling postoperative pain and its safety in comparison with cFNB. The secondary aim of the study was to analyze the ease of use with the STTS system for the administration of analgesics by physicians, nurses and patients, using feedback analysis [19,20].

## 2. Materials and Methods

This was a retrospective observational monocentric study on a representative group of 50 patients (30 (60%) women; 20 (40%) men) treated with TKA at a mean age of 66 (min 22–max 83 years) between October 2018 and June 2020. All the patients were treated with perioperative multimodal analgesia. We compared a group of patients receiving the SSTS postoperative analgesia (SSTS group, 50 patients) versus a control group of patients who were treated according to our standard pain management with continuous femoral nerve block (cFNB group, 21 patients). The inclusion criteria were age >18 years and patients’ ability to describe and report pain. Exclusion criteria were administration of general anesthesia, postoperative admission to the intensive care unit, and previous history of chronic pain. All data used in this study were obtained from the patients’ hospital records and included patient demographic data and all data regarding intraoperative management, postoperative analgesia, and pain scores. All relevant data are within this paper.

### 2.1. Perioperative Management

All patients received IV midazolam as an anxiolytic premedication. Antibiotic prophylaxis was administered with 2 g of cefazolin. In our ERAS protocol, spinal anesthesia was performed with 10 mg of hyperbaric bupivacaine 0.5% without adjuvant or 7.5 mg/mL (from 1.5 to 2 mL) isobaric levobupivacaine. Both groups received subarachnoid anesthesia and 68 patients (95.8%) underwent one shot single femoral nerve block (FNB) (Ropivacaina 0.5% 20 mL one shot). In case of pain not properly controlled (NRS ≥ 4), patients of both groups received a “rescue dose” with ketoprofene 160 mg ev, i.m 2/24 h or toradol 30 mg (max × 2/die); in the case of intolerance to FANS, paracetamol 1 g ev three times a day, Tramadol 50 mg max 3/die or morphine 5 mg subcutaneous were chosen.

After surgery, the patients in the SSTS group received sufentanil 15 μg upon need plus paracetamol (1 g) three times a day. The SSTS in our population was typically prepared by an anesthetist with a mean set up time 2.7 min (±1.8). Conversely, patients treated according to our standard protocol (cFNB group/control) underwent continuous femoral nerve block (cFNB) with continuous infusion of local anesthetic (ropivacaine 0.2% 7 mL/h plus paracetamol 1 g (ev] four times a day). The reasons for choosing the analgesic technique were not completely at the patient’s discretion, but we consider the ability in terms of drug administration device as a discriminating factor. However, the good preliminary results and the controlled self-administration modality have directed the choice of many patients towards this modality of pain management.

The endpoints measured were pain intensity at rest, number of adverse events, number of rescue doses, length of hospital stay, and usability scores. Pain was measured using the numeric rating scale (NRS) (range: 0 = no pain to 10 = worst pain possible). Pain perception during the postoperative period was calculated using the NRS scale at 24 h (T1), 48 h (T2), and 72 h (T3) for both groups. Pain was evaluated in both groups as the mean scores for T1 throughout T3 (T1 to T3 are the three postoperative days). For the SSTS group, pain scores were collected for the day prior to surgery (T-1), the day of surgery at delivery of the SSTS device to the patient (T0), the first administration, and 2-4-8- and 16 h after the first nanotablet. The practicality and manageability of the drug administration device by the staff (nurses) was analyzed through semi-quantitative data (time for preparation, number of doses administered, and subjective evaluation scales) that measure the usability of the device by the staff, and the understanding of how it works by the patient.

### 2.2. Statistical Analysis

All data reported in this study are expressed as the mean, standard deviation (SD), range, and median for each value. Continuous variables were compared between two groups using the two-sample *t*-test and among three groups using one-way analysis of variance (ANOVA). The chi-squared test was used for categorical variables. Pain intensity between the groups was also evaluated using the “U” test (Mann–Whitney), comparing the observation at baseline and final follow-up. Statistical significance was set at a *p*-value of less than 0.05. The data were analyzed using Minitab v.18^®^ (Minitab 18 Statistical Software (2019). Minitab, Inc.: State College, PA, USA).

## 3. Results

The study compared 50 patients in the SSTS drug group and 21 patients in the cFNB group presenting similar characteristics (age, comorbidities, type of surgery) and differing limitedly to the analgesic technique received.

*Pain scores.* In the SSTS group, the NRS scores were: 2.38 (T1), 3.28 (T2), and 2.34 (T3). The NRS scores for the control group were: 3.81 (T1); 3.43 (T2); and 2.57 (T3). The overall differences in pain intensity and the difference in time point T1 between the groups were significantly different (*p* = 0.008) (Figure 2).

The NRS evaluated at T2 and T3 and compared between the two groups was not statistically significant (Figure 3).

### 3.1. Rescue Analgesic Dose and Patient Satisfaction

Patients treated in the SSTS group required significantly less rescue doses than those in cFNB control (5% of SSTS patients vs. 60% patients, of which 25% on T1, 15% on T2, 10% on T3, and 10% on T4). The mean duration of treatment with SSTS from the first to the last nanotablet received was less than 48 h (no patients used STSS for more than 72 h). Rescue doses were required less often in the SSTS group (66% of SSTS patients) than the cFNB control (100% patients), which resulted in fewer calls and satisfaction of the nursing staff (Figure 4).

### 3.2. Adverse Events

Adverse events reported in the SSTS group were postoperative nausea and vomiting (PONV 6%), nausea (10%), vomiting (2%), constipation (24%), bladder globe (4%), somnolence (2%), headache (2%), and hypertension (2%) disorientation (6%) (Table 1). The adverse events in the SSTS group are shown in Table 1.

### 3.3. Hospitalization

Patients were dismissed from the hospital for a mean of 7.06 days in the SSTS group vs. 8.62 of the control group. There was a significant difference in the time to hospitalization, with fewer days in the SSTS group (*p* = 0.039) (Figure 5).

### 3.4. Usability

Most patients confirmed that the SSTS was easy to use and did not report any difficulties in using the device. Most patients report no motor discomfort and 86% of patients reported complete pain relief. Three patients (6%) discontinued SSTS due to ineffective pain relief after T3, and four (8%) due to the malfunctioning of the device (these were excluded from the statistical analysis at the T2 and T3 evaluation).

## 4. Discussion

This study evaluated the efficacy and safety of the SSTS in comparison with a classic multimodal analgesic approach associated with cFNB for pain control in patients treated with TKA. Our results confirmed the efficacy and safety of SSTS in TKA, in agreement with previous studies [22,27,28]. However, we used a different approach than the other studies: Melson et al. compared SSTS to Intravenous patient-controlled analgesia morphine sulfate enrolling patients scheduled for elective major open abdominal or orthopedic (hip or knee replacement) surgery [22]. Jove et al. evaluated the efficacy SSTS 15 μg vs. an identical placebo system for the management of pain after knee or hip arthroplasty [27]. Minkowitz et al. performed another placebo-controlled trial evaluating sufentanil sublingual tablet 30 mcg [28]. This study reflects the clinical experience of a ward in which patients who used the SSTS were rationally selected from a larger group and therefore cannot claim the authority of a randomized trial. These results must be considered evidence for real-life clinical practice.

There are two main limitations of the study: (1) as a consequence of the relative small number of patients, it was not possible to conduct a comparative statistical analysis for all possible variables, but only for the main aspects (pain score, rescue, nursing assistance, adverse effects, and hospitalization) considered in the study; (2) cFNB provided better analgesia compared with single-shot FNB, however, our control group received only a cFNB plus paracetamol whereby it is well-known that it does not cover the whole pain receptions [29,30]. For sufficient pain control, a sciatic nerve block (SNB) would have to be added. Continuous SNB has been reported to improves analgesia and decreases morphine request compared with single-injection sciatic nerve block in patients undergoing TKA, but it influences the possibility of an early rehabilitation in these patients whereas is not observed in cFNB [31]. This is the main reason that limited its use in our cohort in the period of analysis. A recent metanalysis analyzed the analgesic benefits of adding sciatic nerve block (SNB) to FNB following TKA and authors concluded that SNB can seems to reduce postoperative opioid consumption in these patients, even if the available evidence is marked by significant heterogeneity [32]. Other studies provide evidence-based supports to the benefits of SNB as a complement to FNB in TKA [Zorrilla]. Furthermore, the FNB group did not receive planned doses of stronger analgesics, such as retarded opioids, to cover up for the incomplete pain control by the FNB. Thus, it is natural that the FNB group needed more rescue analgesics. This uncontrolled bias of our series may support that the positive analgesic effect with SSTS might not only be due to superior pain control, but also due to an optimizable current standard pain management in the control group.

Few RCTs have specifically addressed the role of SSTS in postoperative pain. We observed a significant difference in postoperative pain intensity between the two groups, with better control using the SSTS protocol, as reported in other studies. Scardino et al. reported lower pain upon movement in the SSTS group (95 patients) than in the cFNB group (87 patients, control group) at all time points, and pain was properly managed when local infiltration anesthesia (LIA) was worn off [33]. The efficacy of the treatment adopted on pain relief was also suggested by the lower request of rescue doses (60% of patients on cFNB vs. 5% of those using SSTS) [33]. Jove et al. showed that STSS 15 mcg could improve the summed pain intensity difference (SPID) 48 h after TKA or total hip arthroplasty (SSTS 76 vs placebo −11) [27]. They also reported a mean NRS pain scores at 24 h of 3.9 (±0.2) for SSTS versus 5.1 (±0.4) for placebo (*p* = 0.002) [27]. However, in both studies, the significant and pronounced effect of SSTS should be analyzed considering that the patients in control group were treated with placebo [27]. Other authors did not find a clinically significant pain improvement analyzing a series of patients treated with TKA in multimodal analgesia under an enhanced recovery protocol, comparing sublingual sufentanil 15 mcg tablet system and oral oxycodone [13]. Different investigators confirmed these results on different scales and at different intervals. We normalized all NRS to a 0–10 range. Most of the authors considered pain intensity between 12 and 24 h after surgery, whereas pain intensity over the first 24 h was reported in 12 studies (including 2327 patients with 1844 in the SSTS group). All participants in SSTS group reported NRS ″4 within 24 h after surgery, [34] and only one trial recorded NRS at 12 h of 5 and at 24 h of 4.5 [2]. It is important to point out that this was the highest pain score recorded among patients treated with SSTS [34]. Numerous studies reported the use of rescue medication medication, IV morphine, oral morphine, oral oxycodone, and acetaminophen, ketorolac) if analgesia with SSTS is insufficient [22,24,27,28,33,34,35,36,37]. In our study, we found a clear reduction in calls for rescue analgesia (*p* < 0.0001), even if it appeared that the pain intensity was not significantly reduced, except for the mean value of NRS in the first 24 h.

Our hypothesis is that SSTS offers a more stable analgesic coverage, distinguishing itself from the strategy used for the controls (ropivacaine + paracetamol), which instead requires continuous rescue intervention to ensure adequate pain control. Furthermore, comparable NRS values but with a higher number of calls suggest that NRS values were not detected every time the patient required rescue intervention but after the therapy had taken effect. This is explained by the fact that in real life, patients complaining of pain are treated without necessarily paying attention to its measurement using appropriate scales. However, we think that a RCT with the combined use of both techniques (SSTS + cFNB) should be carefully evaluated due to promising results. The SSTS system allows the patient to sufficiently control for any pain not being covered by regional analgesia or particularly in case of a failed block, but still, additional regional analgesia may save opioids and thus reduce opioid-related side effects. In addition, a saphenous nerve block may have some advantages over the cFNB regarding motor function and early mobilization.

A clear advantage of SSTS is the administration modality, without the need for additional venous access, which reduces the intrinsic risk of infections linked to invasive routes of administration in selected patients (e.g., oncologic patients). Regarding usability, SSTS was well accepted by both patients and care providers, and Turnbull et al. described SSTS as easy and quick (approximately 4 min per patient), and its functioning is easy to explain to patients. A further advantage of SSTS with sublingual tablets is the oral administration, reducing the risk of infection, analgesic gaps, or conduct obstruction related to the classic pump or IV catheter infusion [27,35,38,39].

We reported an extremely significant reduction in calls to nurses (*p* < 0.0001). This in the SSTS group supports the managerial superiority of the SSTS, both in terms of analgesic therapy (self-administered) and the ward, allowing a redistribution of human resources. In fact, nursing staff are not employed in the preparation and administration of analgesic therapy, which increases patient comfort. The second aspect concerns the intrinsic risk, linked to human factors, of possible mistakes in the administration of analgesic therapy. This risk is eliminated by delegating dosage control of analgesia to the device.

The incidence of adverse effects reported in our study were line with the literature: PONV (6%), with nausea (10%), constipation (24%), disorientation (6%), and bladder globe (4%) the most frequently observed. In a study of 100 patients treated with SSTS after knee or hip arthroplasty, the adverse events reported were nausea (34.9%), vomiting (10.8%), desaturation (7.0%), constipation (4.8%), pruritus (4.8%), headache (4.1%), dizziness (5.1%), and confusion state (2.5%) [27]. Similar results have been reported by Melson et al., with nausea (42.9%), vomiting (13.8%), desaturation (9.6 %), and constipation (11.3%) the most frequent [22]. Although studies have shown that SSTS could save time in the ward, the most interesting advantage we observed from the use of SSTS was the patient’s satisfaction rate, as reported in previous papers [13,21,35,40,41,42].

The mean hospitalization duration was significantly lower in the SSTS group than in the control group (7.06 days vs. 8.62 days; *p* = 0.039). Scardino et al. described how patients in the SSTS group were able to ambulate earlier after surgery and achieve the goals set by the fast-track protocol. Moreover, patients in the SSTS group were all discharged from the hospital within three days of surgery compared to only 36% of the cFNB patients [29]. This study reported similar results but differed from ours in the combined use of cFNB with multimodal drug therapy which included oxycodone/nalaxone 10 mg/5 mg tablets twice daily plus ketoprofen 100 mg 2 capsules/24 h or, in case of NSAID intolerance, paracetamol 1 g three times a day [29].

## 5. Conclusions

The sublingual sufentanil system (SSTS system, Zalviso^®^) is an investigational patient-controlled system that utilizes the SST system to treat moderate-to-severe acute pain in the hospital setting. This system has advantages over IV PCA and has been demonstrated to provide rapid analgesia and achieve high patient and nurse satisfaction ratings in clinical use. Our data confirm the safety and tolerability of SSTS in the management of postoperative pain, resulting in a high level of patient satisfaction and acceptance of pain control. This study demonstrates how the SSTS reduces the need for interventions by healthcare personnel, allowing a redistribution of resources and a reduction in the use of analgesic drugs. A randomized controlled trial is required to provide a conclusive answer. Since this study only evaluated results directly after surgery, future studies could be expanded with long-term results, preoperative pain scores, and the inclusion of more patients could help obtain significant results.

## Figures and Tables

**Figure 1 jcm-11-06864-f001:**
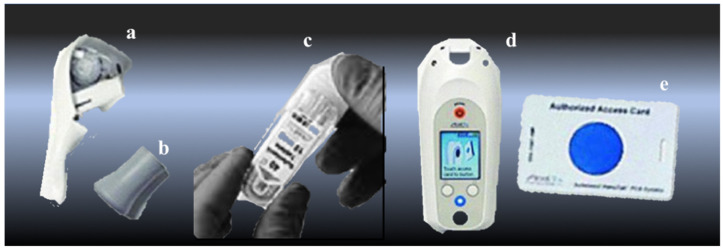
The Zalviso^®^ system consists of a disposable dispenser tip (**a**) and dispenser cap (**b**); a cartridge of Sufentanil sublingual 15 mcg tablets in a disposable bar-coded cartridge (**c**); a reusable handheld controller (**d**) and an authorized access card (**e**).

**Figure 2 jcm-11-06864-f002:**
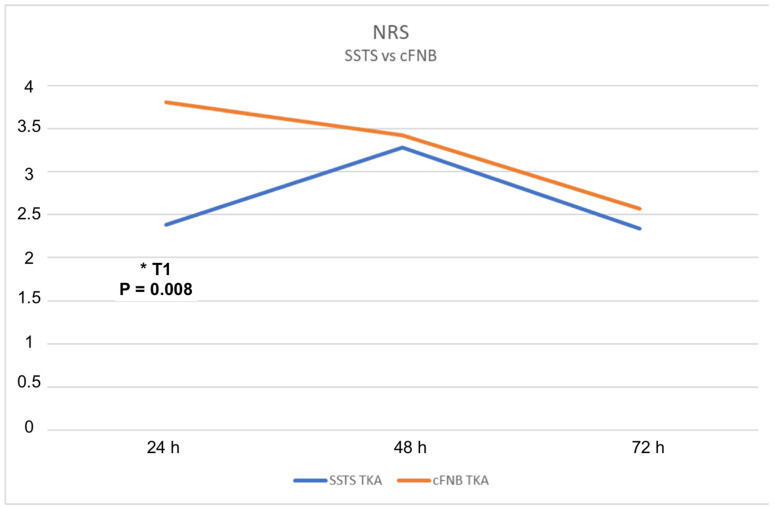
NRS were lower in the SSTS group compared to cFNB group at each time points measured with significant different in T1 (*p* = 0.008) *.

**Figure 3 jcm-11-06864-f003:**
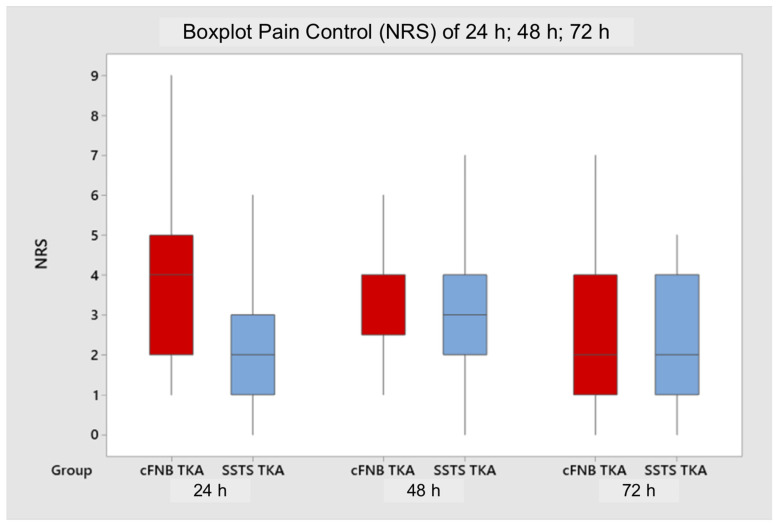
Boxplot analysis of pain control at 24 h, 48 h, and 72 h between the two groups. Horizontal line represents mean value, and the square box represents interquartile range.

**Figure 4 jcm-11-06864-f004:**
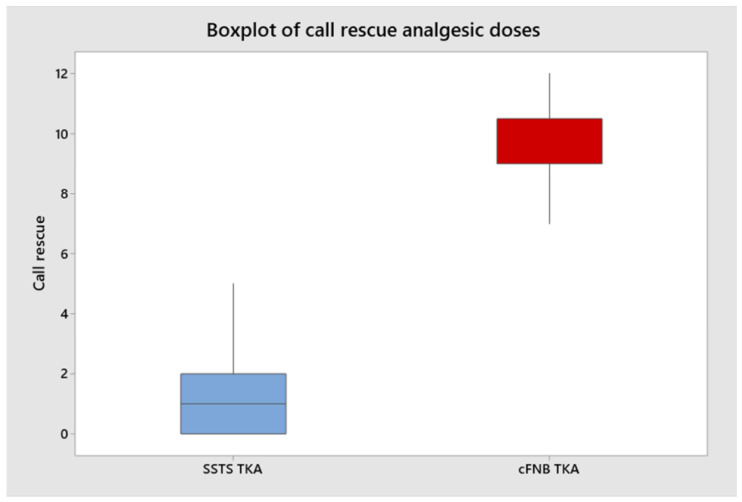
Boxplot analysis rescue doses between the two groups. Horizontal line represents mean value and the square box represents interquartile range.

**Figure 5 jcm-11-06864-f005:**
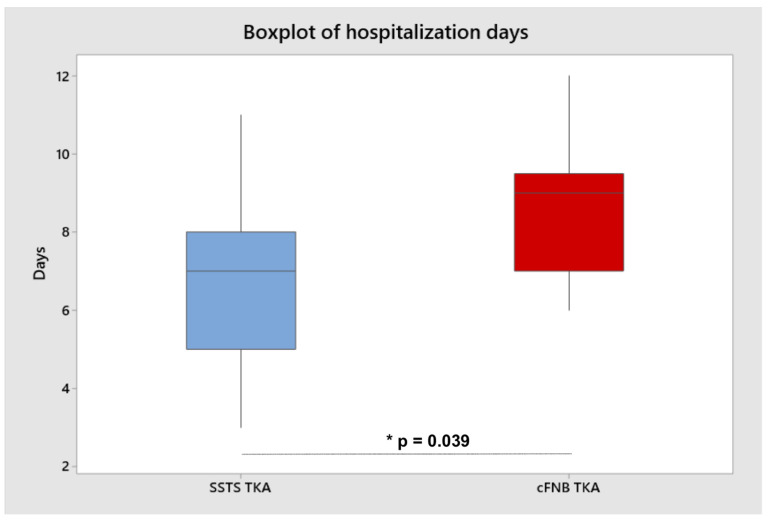
Boxplot analysis hospitalization between the two groups. Horizontal line represents mean value, and the square box represents interquartile range. (*p* = 0.039) *.

**Table 1 jcm-11-06864-t001:** Adverse events reported in the SSTS group vs. cFNB group.

	SSTS (50)	CFNB (21)
PONV	(3) 6%	(3) 15%
Nausea	(5) 10%	(6) 30%
Vomiting	(1) 2%	(3) 15%
constipation	(12) 24%	(4) 20%
Bladder globe	(2) 4%	(1) 5%
somnolence	(1) 2%	(2) 10%
headache	(1) 2%	(1) 5%
hypertension	(1) 2%	(3) 15%
disorientation	(3) 6%	(1) 5%

## Data Availability

The dataset supporting the conclusions of this review is available upon request to the corresponding author.

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
