# Peer review of "Sublingual Sufentanil Tablet System (SSTS-Zalviso®) for Postoperative Analgesia after Orthopedic Surgery: A Retrospective Study"

_jcm, 2022, doi:10.3390/jcm11226864_

Round 1

Reviewer 1 Report

The authors compared pain control with a sufentanil sublingual delivery system (SSST-Zalviso) to continuous femoral nerve blocks for total knee arthroscopies. The authors concluded that the sufentanil sublingual delivery system provided better pain control and 24 hours and reduced the number of rescue interventions. 

There are several points that must be considered:

1. The manuscript would be enhanced by a diagram of the SSST-Zalviso.

2. What I find that is critical is that citation #29 (Scardino, et al) is almost an identical study with the same results (P9, L 249-253). The authors state that their study confirms previous reports (P7, L 178-181).  The authors must justify how their results add new information to the existing literature.

Author Response

Sublingual sufentanil tablet system (SSTS-Zalviso®) for postoperative analgesia after orthopedic surgery: a retrospective study

Reviewer Remarks

Authors’ Responses

Reviewer 1:

Does the introduction provide sufficient background and include all relevant references? Yes

Are all the cited references relevant to the research? Yes

Is the research design appropriate?               Yes

Are the methods adequately described? Yes

Are the results clearly presented? Yes

Are the conclusions supported by the results? Yes

Thank you Reviewer 1. The requested corrections and comments have been highlighted in red.

Reviewer 1.

The authors compared pain control with a sufentanil sublingual delivery system (SSST-Zalviso) to continuous femoral nerve blocks for total knee arthroscopies. The authors concluded that the sufentanil sublingual delivery system provided better pain control and 24 hours and reduced the number of rescue interventions.

There are several points that must be considered:

Thank you Reviewer 1.

1. The manuscript would be enhanced by a diagram of the SSST-Zalviso.

Thank you Reviewer 1 for your comments. We added a new figure 1 and the related captation:

Figure 1:  The Zalviso® system consists of a disposable dispenser tip (a) and dispenser cap (b); a cartridge of Sufentanil sublingual 15 mcg tablets in a disposable bar-coded cartridge (c); a reusable handheld controller (d) and an authorized access card (e)

2. What I find that is critical is that citation #29 (Scardino, et al) is almost an identical study with the same results (P9, L 249-253). The authors state that their study confirms previous reports (P7, L 178-181).  The authors must justify how their results add new information to the existing literature.

Thank you Reviewer 1. Other papers reported on the efficacy of STSS, however the design of their studies was different. We underline the differences.

Moreover, this study reflects the clinical experience of a ward in which patients who used the SSTS were rationally selected from a larger group and therefore cannot claim the authority of a randomized trial. These results must be considered evidence for real-life clinical practice.

Our results confirmed the efficacy and safety of SSTS in TKA, in agreement with previous studies [22,27-28]. However, we used a different approach than the other studies: Melson et al compared SSTS to Intravenous patient-controlled analgesia morphine sulfate enrol-ling patients scheduled for elective major open abdominal or orthopedic (hip or knee replacement) surgery [22]. Jove et al evaluated the efficacy SSTS 15 μg vs an identical placebo system for the management of pain after knee or hip arthroplasty [27]. Minkowitz et al performed another placebo-controlled trial evaluating sufentanil sublingual tablet 30 mcg [28].

This study reported similar results, but differed from ours in the combined use of cFNB with multimodal drug therapy which included oxycodone / nalaxone 10 mg / 5 mg tablets twice daily plus ketoprofen 100 mg 2 capsules / 24 h or, in case of NSAID intolerance, paracetamol. 1 g three times a day [29].

Reviewer 2 Report

Angelini et al. compared the sublingual application of sufentanil with a continuous femoral nerve block for postoperative pain control in patients undergoing total knee arthroplasty. In general, the manuscript is nicely written and most aspects of the study are clearly presented and well-discussed.

However, I am concerned about the validity of the comparison with the control group. The control group received only a femoral nerve block (FNB) plus paracetamol whereby it is well known that the FNB does not cover the whole pain reception after total knee arthroplasty. For sufficient pain control, a sciatic nerve block would have to be added. Furthermore, the FNB group didn’t receive planned doses of stronger analgesics, such as retarded opioids to cover up for the incomplete pain control by the FNB. It is thus natural that the FNB group needed more rescue analgesics. The question thus remains, if the standard pain management is just inferior due to an incomplete regional analgesic technique.

I, therefore, request the authors to discuss this very important point in the discussion section and add that the effect might not only be due to superior pain control with SSTS but also due to an optimizable current standard pain management in the control group.

In case you plan to do an RCT it seems to me that a combination of both techniques seems most promising. The SSTS system allows the patient to sufficiently control for any pain not being covered by regional analgesia or particularly in case of a failed block, but still, additional regional analgesia may save opioids and thus reduce opioid-related side effects. In addition, a saphenous nerve block may have some advantages over the cFNB regarding motor function and early mobilization.

Please find some further comments below:

L85: Inclusion criteria: Please describe the reasons for choosing the analgesic technique. Was this entirely at the patient’s discretion? If so, why are there so many patients receiving the new SSTS system compared to the standard treatment with FNB?

L101-102: “Both 101 groups received subarachnoid anesthesia also some of them underwent femoral nerve block (FNB) (Ropivacaina 0.5% 20 ml one shot).” – The authors should clearly describe how many patients received a single shot or continuous nerve blocks. The current statement doesn’t make clear which group is meant and how many patients received single-shot regional analgesia.

L175-176: “Three patients (6%) discontinued SSTS due to ineffective pain relief and 4 (8%) due to the malfunctioning of the device.” – How did you deal with these patients in your analysis? Did you exclude them? If you excluded those patients from your analysis, you must also make sure that any patients with failed blocks in your control group were excluded. 

Author Response

Reviewer 2:

Does the introduction provide sufficient background and include all relevant references? Yes

Are all the cited references relevant to the research? Yes

Is the research design appropriate?               Yes

Are the methods adequately described? Must be improved

Are the results clearly presented? Yes

Are the conclusions supported by the results? Must be improved

Thank you Reviewer 2.

Reviewer 2.

Angelini et al. compared the sublingual application of sufentanil with a continuous femoral nerve block for postoperative pain control in patients undergoing total knee arthroplasty. In general, the manuscript is nicely written and most aspects of the study are clearly presented and well-discussed.

Thank you Reviewer 2 for your revision.

However, I am concerned about the validity of the comparison with the control group. The control group received only a femoral nerve block (FNB) plus paracetamol whereby it is well known that the FNB does not cover the whole pain reception after total knee arthroplasty. For sufficient pain control, a sciatic nerve block would have to be added. Furthermore, the FNB group didn’t receive planned doses of stronger analgesics, such as retarded opioids to cover up for the incomplete pain control by the FNB. It is thus natural that the FNB group needed more rescue analgesics. The question thus remains, if the standard pain management is just inferior due to an incomplete regional analgesic technique.

I, therefore, request the authors to discuss this very important point in the discussion section and add that the effect might not only be due to superior pain control with SSTS but also due to an optimizable current standard pain management in the control group.

Thank you Reviewer 2. We strongly agree with your comment. We added a specific section in the discussion, using your comment as a guide.

There are two main limitations of the study: 1) as a consequence of the relative small number of patients, it was not possible to carry out a comparative statistical analysis for all possible variables, but only for the main aspects (pain score, rescue, nursing assistance, adverse effects and hospitalization), considered in the study; 2) cFNB provided better analgesia compared with single-shot FNB, however our control group received only a cFNB plus paracetamol whereby it is well known that it does not cover the whole pain receptions [Zorrilla][Chan 2014]. For sufficient pain control, a sciatic nerve block (SNB) would have to be added. Continuous SNB has been reported to improves analgesia and decreases morphine request compared with single-injection sciatic nerve block in patients undergoing TKA, but it influences the possibility of an early rehabilitation in these pa-tients whereas is not observed in cFNB [Beebe 2014]. This is the main reason that limited its use in our cohort in the period of analysis. A recent metanalysis analyzed the analgesic benefits of adding sciatic nerve block (SNB) to FNB following TKA and authors concluded that SNB can seems to reduce postoperative opioid consumption in these patients, even if the available evidence is marked by significant heterogeneity [Abdallah 2016]. Other studies provide evidence-based supports to the benefits of SNB as a complement to FNB in TKA [Zorrilla]. Furthermore, the FNB group didn’t receive planned doses of stronger analgesics, such as retarded opioids to cover up for the incomplete pain control by the FNB. It is thus natural that the FNB group needed more rescue analgesics. This uncontrolled bias of our series may support that the positive analgesic effect with SSTS might not only be due to superior pain control, but also due to an optimizable current standard pain management in the control group.

In case you plan to do an RCT it seems to me that a combination of both techniques seems most promising. The SSTS system allows the patient to sufficiently control for any pain not being covered by regional analgesia or particularly in case of a failed block, but still, additional regional analgesia may save opioids and thus reduce opioid-related side effects. In addition, a saphenous nerve block may have some advantages over the cFNB regarding motor function and early mobilization.

Thank you Reviewer 2. Your expert analysis is really interesting and will significantly improved the quality of our paper. We added your comment in the text.

However, we think that a RCT with the combined use of both techniques (SSTS + cFNB) should be carefully evaluated due to promising results. The SSTS system allows the patient to sufficiently control for any pain not being covered by regional analgesia or par-ticularly in case of a failed block, but still, additional regional analgesia may save opioids and thus reduce opioid-related side effects. In addition, a saphenous nerve block may have some advantages over the cFNB regarding motor function and early mobilization.

L85: Inclusion criteria: Please describe the reasons for choosing the analgesic technique. Was this entirely at the patient’s discretion? If so, why are there so many patients receiving the new SSTS system compared to the standard treatment with FNB?

Thank you Reviewer 2.

The reasons for choosing the analgesic technique was not completely at the patient’s discretion, but we consider the ability in terms of drug administration device as a discriminating factor. However, the good preliminary results and the controlled self-administration modality have directed the choice of many patients towards this modality of pain management.

L101-102: “Both 101 groups received subarachnoid anesthesia also some of them underwent femoral nerve block (FNB) (Ropivacaina 0.5% 20 ml one shot).” – The authors should clearly describe how many patients received a single shot or continuous nerve blocks. The current statement doesn’t make clear which group is meant and how many patients received single-shot regional analgesia.

Thank you Reviewer 2. The sentence was not clear. Most of the patients received a one shot single FNB after subarachnoid anesthesia whereas cFNB was considered as postoperative analgesic modality in 21 patients only.

Both groups received subarachnoid anesthesia and 68 patients (95.8%) underwent one shot single femoral nerve block (FNB) (Ropivacaina 0.5% 20 ml one shot).

L175-176: “Three patients (6%) discontinued SSTS due to ineffective pain relief and 4 (8%) due to the malfunctioning of the device.” – How did you deal with these patients in your analysis? Did you exclude them? If you excluded those patients from your analysis, you must also make sure that any patients with failed blocks in your control group were excluded.

Thank you reviewer 2. Thanks for your comment.

No patients in the control group were excluded from the statistical analysis due to failed block whereas 4 patients in the SSTS were escluded at T2 and T3 evaluation. We modified the text according to these aspects.

Three patients (6%) discontinued SSTS due to ineffective pain relief after T3 and 4 (8%) due to the malfunctioning of the device (these were excluded from the statistical analysis at the T2 and T3 evaluation).

Reviewer 3 Report

The following research article focuses on how SSTS helps reduce pain intensity in the first 24h after surgery, indicating better analgesic coverage and implying reduced interventions from healthcare personnel. 

I have the following list of suggestions that should be implemented:

Line 20: 0,05 should be 0.05, α should be P.

Mann- Whitney, space should be removed

Line 22 (p=0,0568) should be (P = 0.0568)

Line 23 (p<0,0001) should be (P < 0.0001).

This format should be edited throughout the manuscript.

Line 58: μ-receptor-specific agonist should be µ-opioid receptor specific agonist.

Line 71: We shuppose should be “we propose”.

Line 74, 75: Space should be removed.

Line 79: (30 (60%) women; 20 (40%) men) should be [30 (60%) women; 20 (40%) men]

Line 127: p-value should be P-value.

Line 135: 2.38(T1), 3.28(T2) and 2.34(T3). Should have a space in between.

Line 137: (p=0.008) should be reformatted as mentioned above.

In Figure 1 and Figure 4, * should be put inside the figureon the point to show significant difference.

Line 167: 8,62 should be 8.62

Line 168: (p=0,039) should be re-edited as mentioned above.

Line 196: Jove et al. should be Jove et al.

Line 199: (p = 0.002). Please reformat.

Line 201: “Authors” should be “authors”.

Line 213: ) should be removed.

Line 214 and 232: (p <0.0001) whoud be re-formatted.

Line 250: p=0.039 should be reformatted.

In Reference section. All years in Reference section should be inside brackets e.g. (2018) to identify them clearly.

Author Response

Reviewer 3.

Does the introduction provide sufficient background and include all relevant references? Yes

Are all the cited references relevant to the research? Yes

Is the research design appropriate?               Yes

Are the methods adequately described? Yes

Are the results clearly presented? Yes

Are the conclusions supported by the results? Yes

Thank you Reviewer 3. The manuscript has been modified according to your and other Reviewers’ comments

The following research article focuses on how SSTS helps reduce pain intensity in the first 24h after surgery, indicating better analgesic coverage and implying reduced interventions from healthcare personnel.

I have the following list of suggestions that should be implemented:

Line 20: 0,05 should be 0.05, α should be P.

Mann- Whitney, space should be removed

Line 22 (p=0,0568) should be (P = 0.0568)

Line 23 (p<0,0001) should be (P < 0.0001).

This format should be edited throughout the manuscript.

Line 58: μ-receptor-specific agonist should be µ-opioid receptor specific agonist.

Line 71: We shuppose should be “we propose”.

Line 74, 75: Space should be removed.

Line 79: (30 (60%) women; 20 (40%) men) should be [30 (60%) women; 20 (40%) men]

Line 127: p-value should be P-value.

Line 135: 2.38(T1), 3.28(T2) and 2.34(T3). Should have a space in between.

Line 137: (p=0.008) should be reformatted as mentioned above.

Thank you Reviewer 3 for the specific suggestions. We changed the text according to your comments.

In Figure 1 and Figure 4, * should be put inside the figureon the point to show significant difference.

Thank you Reviewer 3. We modify the figure 1 and 4, as well as figure captation.

Line 167: 8,62 should be 8.62

Line 168: (p=0,039) should be re-edited as mentioned above.

Line 196: Jove et al. should be Jove et al.

Line 199: (p = 0.002). Please reformat.

Line 201: “Authors” should be “authors”.

Line 213: ) should be removed.

Line 214 and 232: (p <0.0001) whoud be re-formatted.

Line 250: p=0.039 should be reformatted.

Thank you Reviewer 3. We modified the text.

In Reference section. All years in Reference section should be inside brackets e.g. (2018) to identify them clearly.

Thank you Reviewer 3. The reference section has been checked and modified according to journal style. Year should not be included in round brackets, but in bold.

Round 2

Reviewer 1 Report

The authors  have sufficiently addressed the reviewers' comments.